# Effect of Surface Pre-Reacted Glass Ionomer Containing Dental Sealant on the Inhibition of Enamel Demineralization

**DOI:** 10.3390/jfb13040189

**Published:** 2022-10-14

**Authors:** Yuko Ogawa, Mahmoud Sayed, Noriko Hiraishi, Nadin Al-Haj Husain, Junji Tagami, Mutlu Özcan, Yasushi Shimada

**Affiliations:** 1Department of Cariology and Operative Dentistry, Graduate School of Medical and Dental Sciences, Tokyo Medical and Dental University, 1-5-45 Yushima, Bunkyo-ku, Tokyo 113-8549, Japan; 2Division of Dental Biomaterials, Clinic of Reconstructive Dentistry, Center of Dental Medicine, University of Zürich, 8032 Zürich, Switzerland; 3Department of Reconstructive Dentistry and Gerodontology, School of Dental Medicine, University of Bern, 3010 Bern, Switzerland

**Keywords:** adhesion, dental materials, etching, fluoride, glass ionomer, primer, sealant, silica

## Abstract

The effect of a surface pre-reacted glass ionomer (S-PRG)-containing sealant on the demineralization inhibition and remineralization of intact enamel adjacent to the sealant material was investigated. BeautiSealant (BTS, S-PRG sealant, Shofu), Teeth Mate F-12.0 (TMF, fluoride-releasing sealant, Kuraray Noritake Dental), and an experimental silica-filler sealant were investigated. After pH cycling for 10 days, the enamel surface adjacent to the sealant material was observed using confocal laser microscopy and scanning electron microscopy. The polymerized sealant disks were immersed in a demineralized solution (pH: 4.3) to measure pH change. The enamel specimens with polymerized sealant disks were additionally immersed in demineralized solution, followed by energy-dispersive X-ray spectroscopy. The demineralized area of BTS was significantly smaller than that of TMF and SS (*p* < 0.05). The surfaces adjacent to the sealant of TMF and SS were demineralized, while the surface of BTS was comparatively intact. An increase in pH values were observed in the BTS and TMF groups. Enamel surfaces presented an inhibition of demineralization for BTS and TMF, but not for SS. Fluoride uptake from the polymerized sealant was greater for BTS than for TMF. The S-PRG-containing sealant showed a buffering ability, demineralization inhibition, promotion of remineralization, and it can be advised for clinical applications.

## 1. Introduction

In spite of the fact that the incidence of dental caries has shown decreasing tendencies in recent years, dental caries remains a major health problem throughout the world, affecting both adults and children [1,2]. Fissure demineralization in young permanent teeth is known to occur early after the eruption of teeth and has a high susceptibility to the development of dental caries. The main reason for the development of dental caries in the fissure is its complex anatomical morphology, as it presents ideal prerequisites for food-debris accumulation and bacterial growth. Another reason is the immaturity of the enamel surface immediately after eruption [3] because it presents an area susceptible to demineralization by the acidic by-products of bacteria. Another reason is that the deep areas in the fissure are inaccessible and are therefore difficult to clean completely by mechanical or chemical cleaning in both individuals and professionals in dental practice. For these reasons, fissure sealants are widely used in an attempt to seal the deep areas of the fissures prone to dental caries.

The sealants have been shown to be effective in inhibiting caries’ progression when applied to pits and fissure lesions. There are also a variety of resin-based sealants available for the same purpose. Fluoride-containing sealants are reported to be more effective in inhibiting the development of caries when compared to non-fluoride sealants [4]. In addition, fluoride released from the sealant material also has a remineralizing effect, inhibiting the demineralization of intact enamel adjacent to the sealant material [4]. Therefore, fluoride-containing sealants are recommended to be used in attempts to protect the immature enamel.

Typically, conventional resin-based sealants require an initial phosphoric acid etching of enamel prior to sealant application. The use of phosphoric acid cleans the tooth surface and improves bonding, but at the same time it causes the excessive demineralization of the enamel surface [5]. The ideal sealant to protect the intact enamel adjacent to the material would therefore rely on the remineralization of enamel through fluoride release and on the circumvention of the over-etching of the phosphoric acid.

In this respect, the material-containing surface-reactive pre-reactive glass ionomer (S-PRG) fillers with non-invasive self-etching adhesive properties holds potential as sealants [6,7,8,9,10,11]. S-PRG filler can be incorporated into resin-based materials and used clinically as adhesive resin, sealant materials, or resin-composite restorative materials [12]. The manufacturing process of S-PRG is based on a technology in which fluoroaluminosilicate glass, a powder of glass ionomer cement, is ground and heat-treated, and then treated with polyacrylic acid to form a stable glass ionomer phase on the glass surface. Because a glass ionomer phase is formed on the surface, S-PRG-filler-containing materials not only have the excellent properties of glass ionomer cement, but they also entail strength that is comparable to conventional resin composites [13,14]. This material is a biofunctional material that releases a variety of ions, including fluoride, strontium, borate, aluminum, silicate, and sodium. The released ions act for acid buffering [15], the inhibition of demineralization, promoting remineralization [6,12], and anti-plaque formation [16,17]. Fluoride and strontium form the acid-resistant layer and reinforce tooth structure by reacting on hydroxyapatite to convert it to a fluoride apatite and a strontium apatite [18,19,20]. S-PRG fillers are also known to have the ability to release and repeatedly take up fluoride.

The sealant containing S-PRG fillers is a non-invasive preventive material, and its adhesive performance is not inferior to that of conventional resin-based sealants [21]. One clinical study examined the correlation between complete sealant retention and caries control after 18 months between conventional resin-based sealants and the sealant containing S-PRG filler [22]. The retention rate was found to be higher in conventional resin-based sealants but, in turn, caries control was higher in the sealant containing S-PRG fillers.

The S-PRG-filler-containing sealant with a self-etching primer may be effective in protecting intact enamel from over-etching and preventing caries progression. Therefore, in this study, three different sealant systems were investigated, namely the S-PRG-filler-containing sealant with a self-etching primer, a fluoride-releasing sealant with a phosphoric acid etchant, and an experimental non-fluoride-releasing sealant with a self-etching primer. The null hypotheses tested were that the sealant types would not show a difference in terms of (a) the demineralization and remineralization properties of the enamel surface and (b) the protection of the enamel surface against demineralization due to the buffering effect of the materials.

## 2. Materials and Methods

### 2.1. Sealant Materials

In this experiment, two commercial sealants, BeautiSealant (BTS) (Acknowledgments:) and Teeth Mate F-12.0 (TMF) (Kuraray Noritake Dental, Tokyo, Japan), as well as an experimental silica filler (SS) (provided by Shofu, Kyoto, Japan) were used. TMF was used with a 40% phosphoric etchant as a pre-treatment on the enamel surface, while BTS and SS were used with a self-etching primer according to the manufacturer’s instructions. The material composition and formulation, including the pre-treatment agents used in this study, are listed in Table 1.

### 2.2. Specimen Preparation and pH Cycling

A flow chart of the experimental procedure conducted in this study is shown in Figure 1a. Enamel blocks (9 × 9 × 1.5 mm^3^) from bovine incisors were prepared using a diamond saw (Isomet Buehler, Lake Bluff, IL, USA). The bovine samples were obtained as discarded specimens from authorized procedures approved by the Food Safety Commission of Japan, Ministry of Health, Labor and Welfare. The minimum number of samples were determined by a pilot study and were based on a power calculation of 0.8 with a confidence level of 95%: the number of specimens was set to 12. 

The specimens were polished with 600-, 800-,1200-, 1500-, and 2000-grit silicon carbide papers (Fuji Star, Sankyo Rikagaku, Saitama, Japan) under running water. The debris from the surface was ultrasonically removed for 15 min in distilled water. The marginal area of the polished enamel surface was coated using nail varnish (Revlon Nail Enamel, Revlon, NY, USA), except for the 6 × 6 mm^2^ treatment window, which was pre-treated according to the manufacturer’s instructions for each material. The specimens in BTS and SS were coated with a layer of BeautiSealant Primer for 5 s and air-dried. Afterward, the sealant was applied and photo-polymerized for 10 s. The specimens in the TMF group were treated with phosphoric etchant (K-etchant gel, Kuraray Noritake Dental) for 40 s and rinsed with water. After air-drying, the inner lesion area of 3 × 3 mm^2^ within the pre-treated area was filled using the corresponding sealant and photo-polymerized using an LED light source (Morita Pencure 2000, Morita, Osaka, Japan) for 20 s.

The pH cycle was carried out for 10 days by immersing the specimens in the demineralization solution (0.2 M lactic acid, 3.0 mM CaCl_2_ 1.8 mM KH_2_PO_4_, pH 4.2) [23] for 3 h and in the remineralization solution (0.02 M HEPES, 3.0 mM CaCl_2_, 1.8 mM KH_2_PO_4_, pH 7.0) [23] for 21 h at 37 °C. A volume of 2 mL of demineralization and remineralization solution was used in each cycle. After 10 days of pH cycling, the enamel specimens were rinsed with deionized water (Milli-Q water, Millipore, Meguro-ku, Tokyo).

A vertical section was then made in the middle of the specimen using a diamond saw, and the cross-sectional surface for each section was polished with 600-, 800-, and 1200- grit silicon carbide papers, followed by the use of diamond paste (6, 3, 1, and 0.25 μm, DP-Paste, P, Marumoto Struers, Tokyo, Japan).The polished surface was observed under a confocal laser-scanning microscope (VK-X150, Keyence Corporation, Tokyo, Japan) to analyze the demineralized area of each specimen. In the cross-sectional view, the enamel with an area of 0.5 mm adjacent to the sealant was observed to evaluate the demineralized area. The data were analyzed using an analyzer program (MultiFileAnalyzer V1.3.1.120, Keyence, Osaka, Japan). The enamel surface was sputter-coated with gold and observed using a scanning electron microscope (JSM- IT100, JEOL Ltd., Tokyo, Japan) at an acceleration voltage of 20 kV.

### 2.3. Inhibition of Demineralization

A flow chart of the experimental procedures conducted in this study is shown in Figure 1b. Bovine enamel blocks (5 × 5 × 1.5 mm^3^) were prepared as described above. Nail varnish was applied to the sides and bottom of the enamel block, except for the polished surface. In order to prepare the polymerized sealant disks, each sealant was placed in a silicone mold (diameter: 13 mm, thickness; 1 mm) between two cover glasses, and immediately photo-polymerized on both sides. Two disks of polymerized sealant were placed together with each enamel block and immersed in 5 mL of demineralization solution (0.2 M lactic acid, 3.0 mM CaCl_2_, 1.8 mM KH_2_PO_4_, pH 4.2) [23] and incubated at 37 °C for 10 days. The demineralization solution was not replaced throughout the experimental period. After incubation, the enamel specimens were washed thoroughly with deionized water, dehydrated in a desiccator, and coated with osmium. The morphology of the enamel surface was observed using a field-emission scanning electron microscope (FE-SEM, JSM-7900F; JEOL, Tokyo, Japan) under 15 kV. Energy-dispersive X-ray spectroscopy (EDS, JED-2300; JEOL, Tokyo, Japan) was used to detect the fluorine absorption on the enamel from the polymerized sealant specimens.

### 2.4. pH and Buffering Effect

The polymerized sealant disks were prepared in the same manner as described above. The solution was not replaced throughout the experimental period. pH was measured after 3 h, 6 h, 12 h, 2 days, 3 days, 5 days, 7 days, and 10 days using pH electrodes (LAQUA, HORIBA, Kyoto, Japan).

### 2.5. Statistical Analysis

Shapiro–Wilk tests were used to test the normal distribution of the data. The demineralized area and fluoride detected by EDS were analyzed using two-way analysis of variance (ANOVA) and Tukey’s-HSD post-hoc tests. Statistical analysis was performed using the software package (SPSS, IBM Corporation, NY, USA; Version 23 for Windows). The significance level was set at *p* < 0.05.

## 3. Demineralization after pH Cycling

The CLSM images from the enamel cross-sections after pH cycling are shown in Figure 2. In the BTS group, the enamel surface adjacent to the sealant showed an absence of surface loss (Figure 2A). In the TMF group, the surface pre-treated with a phosphoric acid etchant showed surface loss and irregular concavity (Figure 2B). On the other hand, the advanced surface loss on the enamel surface was present in the SS group (Figure 2C).

The demineralized area enclosed by the dotted line was assigned the demineralization data. Scale bar = 50 μm

Multiple comparisons by ANOVA and Tukey’s test showed significant differences among the three groups. The demineralization data of the specimens is shown in Figure 3. The demineralized area of the BTS group was significantly smaller than that of the SS and TM groups (*p* < 0.05).

The enamel surface adjacent to the sealant material is shown in Figure 4. In the BTS group, the enamel surface was relatively intact, and no demineralized appearance was observed. Instead, small particles and depositions were present. On the other hand, prismatic enamel patterns were observed in the TMF and SS groups, indicating demineralized enamel morphology (Figure 4B,C). This surface texture was prevalent in the SS group compared to the TMF group.

## 4. FE-SEM/EDS Findings after Demineralization

The fluorine content of the enamel surface is shown in Table 2. The mass percentage composition of fluorine was the highest in the BTS group and was significantly higher compared to all other groups (*p* < 0.05). In the SS group, fluorine could not be detected.

TFE-SEM images of enamel surfaces after 10 days are shown in Figure 5. In the BTS group, the enamel surface was covered with enamel smear, and there was no sign of demineralization (Figure 5A). In the TMF group, the enamel surface was similar to the BTS group, showing no signs of demineralization (Figure 5B). However, in the SS group, the demineralization solution removed the enamel smear and exposed the prismatic patterns of the enamel structure (Figure 5C).

## 5. pH Measurement

Figure 6 shows the changes in pH values in the solutions containing the polymerized sealants. The pH of the solution with BTS and TMF increased from pH 4.2 to pH 4.8 and pH 4.6 at 10 days, respectively. The pH of the solution with SS did not change significantly *(p* > 0.05). The pH of the BTS group was significantly higher than that of the TMF and SS groups in all measurement time periods (*p* < 0.05).

## 6. Discussion

For BTS and TMF, the enamel smear produced by polishing remained even after immersion in t In this study, the effect of tested dental sealants on enamel was examined by pH cycling and the demineralization challenging method in an attempt to inhibit demineralization. The pre-treatment manner was also considered using either a phosphoric acid etchant or self-etching primer. Based on the results, and because BTS was more effective in preventing and inhibiting the progression of demineralization after pH cycling, and in the prevention of demineralization, the null hypotheses were rejected.

In this study, using pH cycling, the enamel and demineralization area adjacent to the sealant were measured. The results showed that the depth of demineralization increased in the SS and TMF groups, but almost no demineralization was observed in the BTS group [24]. This finding can be explained by two mechanisms: (i) whether the pre-treatment method had a negative effect on the enamel surface or not, and (ii) how the sealant material inhibited demineralization and promoted remineralization during the pH cycle. In this experiment, the phosphoric acid etchant was used as a pre-treatment for TMF, and the self-etching primer for BTS and SS. TMF is a fluoride-containing sealant, but the enamel surface was demineralized after pH cycling, suggesting that the demineralized enamel induced by the phosphoric acid etchant could not be remineralized. For SS, the enamel surface was treated by the mild self-etching primer. However, the enamel surface was severely demineralized after pH cycling. This outcome emphasizes the fact that the enamel surface was mildly demineralized. Conversely, the SS sealant had no effect on the promotion of the remineralization. Considering the results of the pH cycling, in the case of BTS, pre-treatment had no negative effect, and this sealant promoted the remineralization of the adjacent enamel.

The demineralization challenging revealed that the enamel surface was eroded in the group SS and it had no effect on surface erosion in the BTS and TMF groups. In this method, the enamel specimens were placed in a demineralization solution, along with each polymerized sealant disk. he demineralizing solution for 10 days. This result indicated that the polymerized disks of BTS and TMF buffered the acidity of the solution and/or released ions that were effective in preventing demineralization. Furthermore, the buffering effect of the BTS and TMF group was confirmed by the pH measurement results: the pH of the demineralization solution was above 4.8 for the BTS group and 4.6 for the TMF group. In contrast, for the SS group, the pH was below 4.4. This difference in pH indicates that the enamel surface was protected by the BTS and TMF sealants, not showing erosion.

The deposition or precipitation on the surface of the BTS group can be described as a remineralization phenomenon resulting from pH cycling. Remineralization is defined as the process of the formation of hydroxyapatite crystals on the enamel surface, due to the supply of calcium and phosphate ions [25]. In the pH cycling study, the enamel surface released calcium and phosphate ions during the demineralization cycle at pH 4.2. These ions were utilized in the remineralization cycle at pH 7.0, which may have resulted in mineral precipitation. This mechanism is facilitated by the presence of fluoride ion [25]. When fluoride ions, as well as calcium and phosphate ions, are present, the formation of fluoroapatite or, more accurately, fluorohydroxyapatite, is conceivable [25]. Therefore, the precipitation on the enamel surface of the BTS group might be related to fluoride ions released from the sealant. The results of EDS analysis showed that fluorine was mostly detected on the enamel surface when the polymerized sealant of BTS was immersed in demineralization solution, followed by the polymerized sealant of TMF, while it was barely detected with the sealant of SS. Thus, the difference in fluoride ions released from the polymerized sealants was related to the presence or absence of deposition on the enamel surfaces. In fact, the continuous and sustained release of fluoride ions from dental fluoride-releasing sealants is expected to prevent secondary caries [26]. As fluoride sources, TMF is based on methacryloyl fluoride methacrylate (MF-MMA) and BTS on S-PRG fillers. Fluoride released from fluoride-releasing sealants could be affected by various intrinsic factors, such as their composition, solubility, and porosity [27,28,29]. MF-MMA is formed by methacryloyl fluoride that is covalently bound to methyl methacrylate [29]. The fluoride ions are slowly released from the MF-MMA compound into the aqueous solution through the hydrolysis of the resin matrix [29]. In the case of BTS, on the other hand, fluoride ions released from the stable glass ionomer phase on the glass surface. In that respect, S-PRG fillers have been reported to release slightly more fluoride than fluoride-releasing resin composites [30]. The present experiment indicated that the fluoride ion uptake was higher in BTS compared to TMF. This result is consistent with a previous report, where BTS demonstrated a high fluoride-recharge capacity [27].

The presence of a glass ionomer phase surrounding the glass core of S-PRG fillers is supposed to release Sr, B, F, Al, and Na ions [31]. Sr ions are essential for tooth mineralization [32,33] and have been reported to promote enamel remineralization in combination with fluoride [34]; meanwhile, B, F, and Sr are considered to have antibacterial effects [35]. Na is present in the composition of S-PRG fillers and is released in conjunction with Al [36]. Moreover, the release of Al promotes the release of F and the formation of an aluminofluoro complex [37].

Multiple ions are thought to be involved in the buffering capacity of S-PRG. Hiraishi et al. focused on the possibility that borate ions released from S-PRG are used to neutralize acidity [15]. In that study, NMR measurements explained that the S-PRG solutions contain mainly aqueous B(OH)_3_, and negatively charged B(OH)_4_^−^ ions were adsorbed on the enamel and dentin surfaces [15]. When tetrahedral boron is converted to trihedral boron, acid is consumed, and the pH consequently increases. The authors pointed out that buffering capacity may be due to the ability of borate solutions to consume hydrogen ions (H^+^) [15,38]. In addition to boric acid ions, ions such as Na^+^, Sr^2+^, and F^−^ are thought to be responsible for the buffering capacity of S-PRG. During the ion exchange process, H^+^ ions, Na^+^, and Sr^2+^ ions are released into the solution to form alkaline compounds, such as Sr(OH)_2_ and NaOH, which buffer the solution [39]. The ion exchange between the fluoride and H^+^ ions in the solution renders the solution into a more alkaline one. Additionally, fluoride ions can induce the precipitation of apatites from acidic precursors, such as amorphous calcium phosphate and octacalcium phosphate [40].

The effect of S-PRG fillers was confirmed by the results of pH cycling in the BTS and SS groups. The enamel specimens of the BTS and SS groups were pre-treated by the same self-etching primer, not causing over-etching of the enamel surface. After pH cycling, the enamel surface around the sealant was not demineralized for the BTS group, but it was severely demineralized for the SS group. This may be attributed to the fact that BTS released various ions, such as fluorine, but SS did not [41,42]. In this experiment, fluorine was also detected in TMF, but demineralized enamel was morphologically evident in SEM images. In the BTS group, however, demineralized enamel was not observed by SEM. Thus, the results suggest that the sealant with S-PRG fillers inhibits the enamel demineralization and promotes remineralization due to fluoride and other ions.

Pit and fissure sealants are applied to the fissure surfaces of teeth that are highly susceptible to dental caries and are difficult to approach with other treatment methods, such as fluoride or mechanical plaque control. The complex structure of the pit and fissure makes it difficult for sealant materials to completely penetrate the fissure. Therefore, etching by phosphoric acid prior to sealant application improves the retention of the sealant, but demineralization may induce the development of caries on over-etched surfaces of young permanent teeth. The application of BTS does not necessitate phosphoric acid pre-treatment. The benefits of BTS are related to the use of a mild self-etching primer and the functional effects of the S-PRG filler to protect the enamel structure.

However, it is difficult to predict the prognosis of sealants as it is influenced by various factors, such as the oral environment, retention to tooth fissure, tooth preparation method, eruption status of the tooth, and the amount of fluoride released. One limitation of this study was that bovine teeth were used instead of human teeth, and the duration of the experiment was limited. Therefore, further investigation is necessary, taking into account various factors, such as the eruption state of the teeth and the oral environment.

## 7. Conclusions

The results of this study indicate that the application of BTS does not require phosphoric acid pre-treatment to improve the retention of the sealant. BTS was more effective in reducing demineralization of the surrounding enamel than TMF. S-PRG fillers inhibit the enamel demineralization and promote remineralization due to ions, such as fluoride, in their composition.

## Figures and Tables

**Figure 1 jfb-13-00189-f001:**
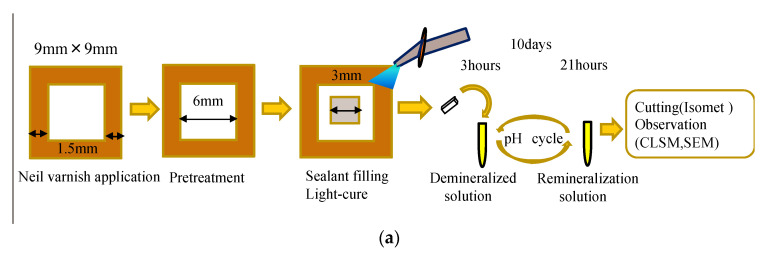
Specimen preparation. (**a**) pH cycling and (**b**) demineralization measurement.

**Figure 2 jfb-13-00189-f002:**
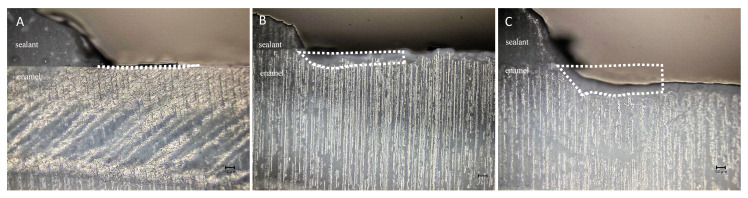
Confocal laser microscope observation of the cross-sections of enamel disks after 10 days pH cycling. (**A**) BeautiSealant (S-PRG filler) (BTS); (**B**) Teethmate F-1 2.0(TMF); (**C**) silica-filler sealant (SS). BTS and SS clearly show demineralization on the enamel surface.

**Figure 3 jfb-13-00189-f003:**
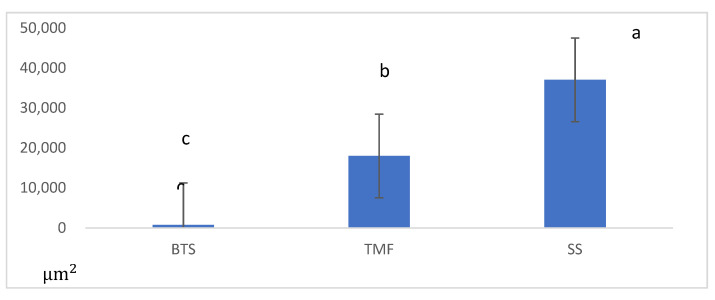
Results of demineralization area after 10 days pH cycling in the different groups. BeautiSealant (BTS), Teethmate F-1 2.0(TMF), silica-filler sealant (SS). Different letters denote a statistically significant difference compared with the other groups (*p* < 0.05, *n* = 12).

**Figure 4 jfb-13-00189-f004:**
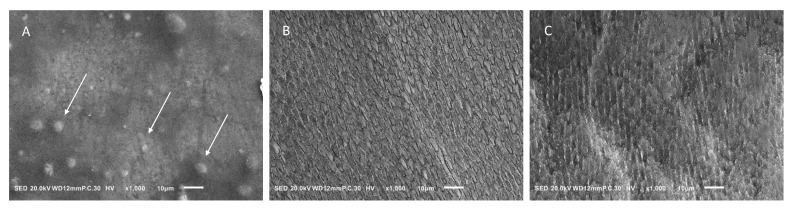
Scanning electron microscopy observation of the surfaces of enamel disks after 10 days pH cycling. (**A**) enamel sample with BeautiSealant (S-PRG filler) (BTS), (**B**) enamel sample with Teethmate F-1 (TMF), (**C**) enamel sample with silica-filler sealant (SS). Particles are present on the enamel surface of BTS, as indicated by the arrows. Magnification: 1000× scale bar = 10 μm.

**Figure 5 jfb-13-00189-f005:**
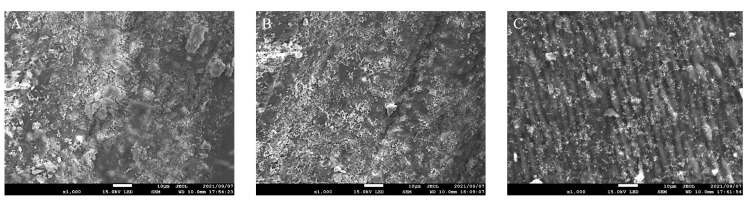
Scanning electron microscopy observation of enamel surfaces after storage in demineralization solution for 10 days. (**A**) BeautiSealant (S-PRG filler) (BTS), (**B**) Teethmate F-1 2.0 (TMF), (**C**) silica-filler sealant (SS). In the BTS and TMF groups, the surfaces were covered with enamel smears and there was no sign of the prismatic outline. In the SS group, there was a pattern of prismatic structure. Original magnification: 1000× scale bar = 10 μm.

**Figure 6 jfb-13-00189-f006:**
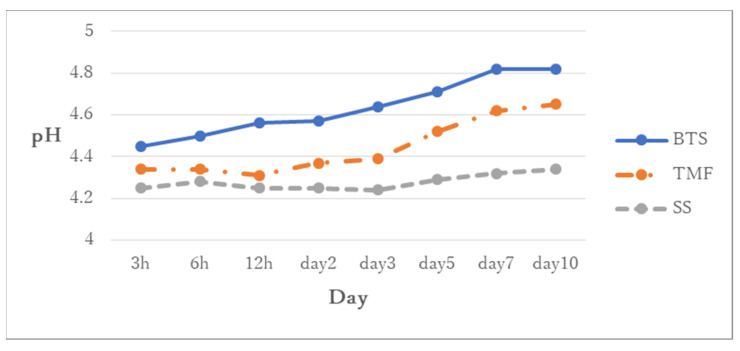
Changes in pH values in the BeautiSealant (BTS), Teethmate F-1 2.0 (TMF), silica-filler sealant (SS). The pH of the solution with the BTS group was always higher than those of the other three groups (*p* < 0.05).

**Table 1 jfb-13-00189-t001:** Brands, abbreviations, chemical composition and manufacturers of the materials used in the study. HEMA, 2-hydroxyethyl methacrylate; MDP, 10-methacryloyloxydecyl dihydrogen phosphate; MF-MMA, methacryloyl fluoride methacrylate; TEGDMA, triethylene glycol dimethacrylate; UDMA, urethane dimethacrylate.

Brand	Abbreviations	Batch Number	Chemical Composition	Manufacturer
BeautiSealant	BTS	042157	UDMA, TEGDMA S-PRG filler	Shofu, Kyoto, Japan
BeautiSealant Primer		071954	acetone, distilled water, carboxylic acid monomer, phosphonic acid monomers and others	Shofu, Kyoto, Japan
TeethmateF-1 2.0	TMF	2H0048	MDP, MF-MMA, TEGDMA, HEMA	Kuraray Noritake Dental, Tokyo, Japan
K-etchant GEL		1N0122	Phosphoric acid, water, colloidal silica, dye	Kuraray Noritake Dental. Tokyo, Japan
Silica-filler Sealant	SS	210323-s3	UDMA, TEGDMA, silica filler	Shofu, Kyoto, Japan

**Table 2 jfb-13-00189-t002:** Mass percentage of fluoride intake of enamel specimens for each sealant material. Different superscript letters in one column indicate significant differences between materials (ANOVA, Tukey’s test). For group abbreviations, see Table 1. Different letters denote a statistically significant difference compared with the other groups.

Sealant Material	[F] Mass (%) (Mean ± SD)
**BTS**	4.7 ± 0.2 ^a^
**TMF**	2.0 ± 0.2 ^b^
**SS**	0.0 ± 0.0 ^c^

## Data Availability

The data presented in this study are available on request from the corresponding author.

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
