# Peer review of "Effect of Surface Pre-Reacted Glass Ionomer Containing Dental Sealant on the Inhibition of Enamel Demineralization"

_jfb, 2022, doi:10.3390/jfb13040189_

Round 1
Reviewer 1 Report
Dear authors,
The paper: Effect of Surface Pre-reacted Glass Ionomer Containing Dental Sealant on the Inhibition of Enamel Demineralization, is of interest and relevant for the journal and scientific community, however, I have some comments.
Best regards

Author Response
Reviewer comment 1:
What is S-PRG filler in the chemical composition of BeautiSealant?
[Author Response]
We appreciate your comments and the opportunity to clarify this important point.
Surface pre-reacted glass-ionomer fillers, which are formed by an acid-base reaction between fluoroboroaluminosilicate glass and polyacrylic acid in the presence of water. The full detail was shown in Patent information: Nakatsuka T, Yasuda Y, Kimoto K, Mizuno M, Negoro N. DENTAL FILLERS. United States Patent No.6,620,861;2003
US6620861B1 - Dental fillers - Google Patents
Reviewer comment 2:
Line 116-117: The enamel surface was polished using silicon carbide polishing
papers (Fuji Star, Sankyo Rikagaku, Japan) up to a final 2000-grit.
- What equipment is used for polishing? What is the sequence of the granulometry of polishing paper? The polishing was done dry or what solution was used?
[Author Response]
Thank you very much for the comment. The specimens were polished with 600-, 800-,1200-, 1500- and 2000-grit silicon carbide papers (Fuji Star, Sankyo Rikagaku, Saitama, Japan) under running water. The manuscript has been revised.
Reviewer comment 3:
Line 136:”...the cross-sectional surface for each section was polished using diamond pastes up to 0.25 μm particle size.”
- What equipment is used for polishing? Number of rotations etc...
[Author Response]
Thank you very much for the comment. The specimens were polished with 600-, 800- and 1,200- grit silicon carbide papers followed by using diamond paste (6, 3, 1, and 0.25 μm, DP-Paste, P, Marumoto Struers, Tokyo, Japan). The manuscript has been revised and details added.
Reviewer comment 4:
Páge 5: Figure 2.
- What is the ampliation used in the observation of the samples?
[Author Response]
The version of CLSM was incorrectly mentioned. A more updated VK-X150 was used in this study. This part has been corrected.
Re: the ampliation:
Photomultiplier tubes (PMTs) used as light receiving elements for lasers are characterized by high sensitivity, fast response, and low noise. This makes PMTs superior to conventional CCD (charge-coupled device) type laser microscopes in the following respects. Measurement is possible even on objects made of materials with low surface reflectance. Measurement is also possible even at large inclination angles. Other benefits are high speed scanning, shortening the measurement time and better optimum contrast
Reviewer 2 Report
The present study addresses the effect of surface pre-reacted glass ionomer containing dental sealant on the inhibition of enamel demineralization
I suggest the authors to add in the Title the study type. The Abstract and Keywords look fine. And the Introduction provide a good overview, rationale and aim. Regarding the methodology I wonder why a sample of 12 specimens was used, did the authors conduct any type of sample size? I suggest the authors to place Figure 1 only after it was mentioned in the manuscript body. In table 1 the manufacturer column should have also the city and country. The statistical analysis sub-heading should be improved: the power analysis should appear before this, in the place the sample size is mentioned in the text, and the power calculation requires more than just the power and confidence, which variable was taken into consideration and what was the effect size and SD? Why did the authors perform two test for normal distribution test? Study strength should be analyzed in the Discussion and the external validity should be debated. At last the reference list do not follow the journal guidelines.
Author Response
Reviewer comment 1:
I suggest the authors to add in the Title the study type.
[Author Response]
We thank for the opportunity to clarify this important point. The title has been revised as “Effect of Surface Pre-reacted Glass Ionomer Containing Dental Sealant on the Inhibition of Enamel Demineralization: An In Vitro Study”.
Reviewer comment 2:
Regarding the methodology I wonder why a sample of 12 specimens was used, did the authors conduct any type of sample size?
[Author Response]
It is important to know the appropriate sample size in the preliminary stages of the experiment. Therefore, a pre-analysis was conducted prior to the experiment to determine the sample size. The minimum numbers of specimens required for quantitative testing were estimated based on a power calculation of 0.8 with a confidence level of 95%. The mean values and standard deviations (SD) used for estimation were derived from the pilot study work. Three samples in each group were tested. As a result, the total sample size required was calculated.
In addition, we referred to the sample size provided our colleagues using bovine enamel and dental adhesive materials. We also referred to the similar method and the sample size used in reference #9 (Kaga M et.al., Inhibition of enamel demineralization by buffering effect of S-PRG filler-containing dental sealant. Euro J Oral Sci. 2014;122: 78-83). These factors were taken into consideration when determining the sample size.
Reviewer comment 3:
I suggest the authors to place Figure 1 only after it was mentioned in the manuscript body. In table 1 the manufacturer column should have also the city and country.
[Author Response]
We thank the reviewers for their suggestions. As being indicated, we have revised Figure 1 to properly place it in the manuscript. We have also added the name of the city and country to the Manufacturer column of Table 1.
Reviewer comment 4:
The statistical analysis sub-heading should be improved: the power analysis should
appear before this, in the place the sample size is mentioned in the text, and the power calculation requires more than just the power and confidence, which variable was taken into consideration and what was the effect size and SD?
[Author Response]
Thank you for providing these insights. The manuscript has been revised.
Besides the ppower calculation, it is also important to determine the sample size by considering the previous experiment that used dental adhesives, bovine teeth and the same analyzing methods.
Reviewer comment 5:
Why did the authors perform two test for normal distribution test?
[Author Response]
Given the number of data under consideration, the Shapiro-Wilk test is a reasonable way to test the normal distribution of the data. The manuscript has been revised.
Reviewer comment 6:
Study strength should be analyzed in the Discussion and the external validity should be debated.
[Author Response]
Thank you for this comment. The discussion has been revised to highlight the clinical significance of this study as well as limitation.
Reviewer comment 7:
At last the reference list do not follow the journal guidelines.
[Author Response]
We have also carefully corrected the references.
Round 2
Reviewer 2 Report
Dear authors, I have no further concern. Thank you